# Emerging Mutual Regulatory Roles between m^6^A Modification and microRNAs

**DOI:** 10.3390/ijms24010773

**Published:** 2023-01-01

**Authors:** Zongqin Mei, Yahao Mou, Nan Zhang, Xiaoyu Liu, Zuoshun He, Shiyan Gu

**Affiliations:** Institute of Preventive Medicine, School of Public Health, Dali University, No. 22 Wanhua Road, Dali 671000, China

**Keywords:** N^6^-metyladenosine, methyltransferases, demethylases, methyl-binding proteins, miRNAs

## Abstract

N^6^-metyladenosine (m^6^A), one of the most common RNA methylation modifications in mammals, has attracted extensive attentions owing to its regulatory roles in a variety of physiological and pathological processes. As a reversible epigenetic modification on RNAs, m^6^A is dynamically mediated by the functional interplay among the regulatory proteins of methyltransferases, demethylases and methyl-binding proteins. In recent years, it has become increasingly clear that m^6^A modification is associated with the production and function of microRNAs (miRNAs). In this review, we summarize the specific kinds of m^6^A modification methyltransferases, demethylases and methyl-binding proteins. In particular, we focus on describing the roles of m^6^A modification and its regulatory proteins in the production and function of miRNAs in a variety of pathological and physiological processes. More importantly, we further discuss the mediating mechanisms of miRNAs in m^6^A modification and its regulatory proteins during the occurrence and development of various diseases.

## 1. Introduction

N^6^-metyladenosine (m^6^A) modification refers to the methylation that occurs at the 6th nitrogen atom of adenine. Since firstly found in 1974, m^6^A has received extensive attention for regulating abundant internal modifications on various RNAs, including mRNAs, microRNAs (miRNAs), small nuclear RNAs (snRNAs), long non-coding RNAs (lncRNAs) and circular RNAs (circRNAs) [1,2,3]. As we all know, the formation of m^6^A modification is dynamically catalyzed by methyltransferases and demethylases, also called “writers” and “erasers”, respectively. The binding proteins, named “readers”, can specifically combine with m^6^A modification and mediate m^6^A biological function in different pathological and physiological processes [4]. At present, m^6^A modification has been widely recognized as a reversible and dynamic epigenetic modification on various RNAs [5], and it is becoming increasingly clear that m^6^A potentially contributes to the occurrence and development of multiple diseases through altering RNA expressions or RNA functions [6]. Recently, the clinical value of m^6^A modification in diseases has become apparent, and m^6^A modification has been commonly utilized as a promising biomarker to diagnose, prevent and treat diseases [7]. 

As the most common non-coding RNAs, miRNAs exert their biological functions through interaction with RNAs or proteins [8]. Recently, emerging works of literature have demonstrated that miRNAs were modified by various chemical modifications, such as m^6^A, m^1^C, m^5^C and m^7^G, which affect the processing and functions of corresponding miRNAs [9,10,11]. Among these modifications, m^6^A attracted the most attention. As reported previous study, m^6^A modification is preferentially concentrated near the 3′-UTR, of which 67% contain ncRNAs, such as the binding sites of miRNAs, implicating that m^6^A and ncRNAs may jointly regulate target mRNAs through cooperation or competition [12]. The latest studies have found that m^6^A modification affects the cleavage, transport, stability and degradation of corresponding miRNAs and influences the interactions between miRNAs and long non-coding RNAs or proteins [13,14,15]. Interestingly, it is becoming increasingly clear that miRNAs also have critical roles in regulating m^6^A modification by changing the regulatory proteins of m^6^A modification [16]. Although some reviews summarize the interaction between m^6^A modification and miRNAs in cancer [17,18,19], studies on m^6^A modification and miRNAs are growing explosively, and their interactive effects are not limited to tumors. Thus, in this review, we summarize methyltransferases, demethylases and methyl-binding proteins which regulate m^6^A modification on RNAs. Subsequently, we sum up the essential roles of m^6^A modification on miRNAs in a variety of pathological and physiological processes. Lastly, we describe the mediation of miRNAs in m^6^A modification and discuss the interactive effects between m^6^A modification and miRNAs during the occurrence and development of the diseases.

## 2. Dynamic Regulation of m^6^A Modification

In recent years, more and more studies have found that m^6^A modification on RNAs can be dynamically regulated by dedicated methyltransferases, demethylases or methyl-binding proteins. A set of these regulatory proteins have been summarized in Figure 1.

### 2.1. “Writers”-m^6^A Methyltransferases

As shown in Figure 1, the addition of m^6^A modification to RNAs is executed by several components. Methyltransferase like 3 (METTL3), the first identified enzyme which catalyzes the m^6^A formation, was isolated and purified from Hela cells in 1994 [20]. In December 2013, Chuan’s group found methyltransferase like 14 (METTL14) has 43% homology with METTL3, and further exploring evidence demonstrated that METTL3 and METTL14 could form stable heterogeneous dimer to catalyze the formation of m^6^A modification [21]. In detail, METTL3 is the sole catalytic subunit, whereas METTL14 is essential for m^6^A deposition by stabilizing METTL3 conformation and recognizing substrate RNAs. The following studies showed that Wilms’ tumor-associated protein 1 (WTAP) was also an essential component for the formation of m^6^A [22]. Specifically, although WTAP has no methyltransferase activity, it acts as an adaptor protein to guide the METTL3-METTL14 heterodimers to co-locate on the nuclear speckle [23]. Interestingly, other adaptor proteins include vir like m^6^A methyltransferase associated (VIRMA, also named as KIAA1429) [24] and Cbl Proto-Oncogene Like 1(HAKAI, also known as CBLL1) [25], and zinc finger CCCH-type containing 13(ZC3H13/flacc) [26] can interact with WTAP and affect the level of m^6^A modification. In addition, lowering the expression of connexins such as RNA binding motif protein 15 (RBM15) and its para family like homologous protein RBM15B can also significantly reduce the level of m^6^A modification [27], suggesting that connexins are also an essential part of the formation of m^6^A modified methyltransferase 16(METTL16) [28]. Methyltransferases like 5(METTL5) [29] and zinc finger CCHC-type containing 4 (ZCCHC4) [30] have also been identified as m^6^A methyltransferases which individually catalyze m^6^A modification on specific structured RNAs (U6 snRNA, 28S rRNA, and 18S rRNA)

### 2.2. “Erasers”-m^6^A Demethylases

Demethylases, known as “erasers”, can selectively remove m^6^A marks from target RNAs and ensure that m^6^A methylation is a dynamic and reversible process. As far as we know, all the m^6^A demethylases currently identified belong to homologous proteins of Escherichia coli dioxygenase ALKB family: Fat mass and obesity associated protein (FTO) and AlkB family of nonheme Fe (II)/α-ketoglutarate (α-KG)-dependent dioxygenases 5 (ALKBH5) and ALKBH3 [31,32]. Initially, FTO was thought to be involved in the occurrence and development of obesity by regulating the expression of genes related to energy metabolism through demethylation [33]. In 2011, a study indicated that the bases which can be demethylated by FTO mainly contain m^6^A modification. Further exploration revealed that the m^6^A modification on mRNAs can be enhanced by siRNA-mediated knock-down of FTO expression while the upregulation of FTO protein inhibits m^6^A level [34]. This is the first confirmation that FTO can remove m^6^A methylation and the first report indicating that m^6^A modification is dynamic and reversible methylation. So far, FTO has been widely recognized as an m^6^A demethylase in a variety of RNAs from human or animal tissues [35]. As another “eraser” of m^6^A modification, ALKBH5 can oxidatively remove m^6^A modification on RNAs in vitro and in vivo [36]. Naturally, ALKBH5 is involved in the mRNA output, RNA metabolism and the assembly of mRNA processing factors in nuclear speckles [37]. It has been reported that depletion of ALKBH3 can effectively increase the m^1^A and m^3^C, suggesting that ALKBH3 is also a demethylase [38,39]. Interestingly, in addition to m^1^A and m^3^C modification, ALKBH3 can demethylate m^6^A modification in tRNA, and ALKBH3-interacted tRNA can increase protein efficiency [40]. At present, although FTO, ALKBH5 and ALKBH3 are homologous protetranslationins, they erase m^6^A marks on RNAs in an independent manner.

### 2.3. “Readers”-m^6^A Methyl-Binding Proteins

Similar to DNA methylation, the biological function of m^6^A modification on RNAs is performed by specific methylated proteins. Recent studies have indicated that m^6^A methyl-binding proteins are a class of protein family, which contains the YTH domain. As early as 2012, studies characterized YTH domain family protein 2 (YTHDF2), YTHDF3 and YTHDC1 as m^6^A binding proteins through the combination of RNA pull down and mass spectrometry [41]. Subsequently, the other YTH domain family protein members, YTHDF1 and YTHDC2, were also confirmed to bind with m^6^A modification [42,43]. More importantly, further functional studies showed that YTHDF1, YTHDF2 and YTHDF3 could mediate the degradation and translation while YTHDC1 and YTHDC2 were related to the splicing and nucleation of target mRNAs which were modified by m^6^A modification [44]. YTHDC1 has been reported to bind ncRNAs rather than mRNAs. [45]. In addition, Human heterogeneous nuclear ribonucleoprotein (HNRNP) A2/B1 can bind to miRNA through m^6^A modification and affect the splicing and transshipment of corresponding miRNA precursor, so as to regulate the miRNA generation and maturation [46]. Eukaryotic initiation factor 3 (EIF3) also has been identified as an m^6^A methylated binding protein of mRNA at 5′-UTR and involved in cap-independent translation [47]. ATP Binding Cassette Subfamily F Member 1 (ABCF1) has been evidenced as a critical mediator of m^6^A-promoted translation under stress and physiological conditions and further exploring indicated ABCF1-sensitive transcripts largely overlap with METTL3-dependent mRNA targets, supporting that ABCF1 may act as an m^6^A methyl-binding protein [48]. In 2018, it was shown that insulin-like growth factor 2 mRNA-binding proteins 1/2/3 (IGFBP1/2/3) were also methyl-binding proteins of m^6^A modification, which could increase the stability and translation efficiency of target mRNAs [49]. In addition to the proteins directly bound to the m^6^A modification mentioned above, some proteins can indirectly bind to m^6^A modification. As we all know, once the methylation modification occurs on RNAs, the conformation of RNA will be changed, and some binding sites will be uncovered, thus leading to the recognition by RNA binding proteins. This phenomenon is vividly known as ‘m^6^A switch’ [50]. For now, two types of m^6^A indirect binding proteins have been found, namely, HNRNPC (heterogeneous nuclear ribonucleoprotein C) and HNRNPG, which involve in transcription and maturation of mRNAs [51,52,53]. Furthermore, FMR1 (Fragile X Mental Retardation 1) is a sequence-context-dependent m^6^A reader, which can interact with m^6^A by depending on RNA secondary structure to inhibit its translation [54].

## 3. Mutual Regulatory Mechanisms between m^6^A Modification and miRNAs

Recently, a large number of studies have demonstrated that m^6^A modification on miRNAs play essential roles in various pathophysiological processes. Most interestingly, emerging evidence revealed that miRNAs also regulate m^6^A modification by altering expressions of m^6^A regulatory proteins [51,55]. So far, mutual regulatory mechanisms between m^6^A modification and miRNAs in multiple diseases have attracted a huge amount of attention. 

As a series of noncoding and single-stranded small molecular RNA with a length of 18–24 nucleotides, miRNAs could target specific mRNA sites and promote degradation or inhibit translation of mRNA [56]. Although there are nearly 3000 miRNAs in mammals, the generation processes of different miRNAs are almost consistent. In detail, upon transcription from DNA, primary transcripts of miRNAs (pri-miRNAs) are spliced by RNase Ⅲ structure domain proteins and double-stranded RNA binding protein Drosha and Di George Syndrome critical region 8 (DGCR8) to form precursor of miRNAs (pre-miRNAs) in the nucleus. Then, pre-miRNAs are exported from nucleus into cytoplasm by forming a complex with a transporter protein exportin-5 and a GTP-binding nuclear protein Ran-GTP [53,54]. Once transported out of nucleus, pre-miRNAs are cleaved into mature miRNAs through another type III RNase Dicer. The mature miRNAs subsequently bind to mRNAs with the help of Ago proteins, thereby affecting the levels or translation processes of corresponding mRNAs [57]. Interestingly, advances in m^6^A modification in recent years have widely broadened mechanisms underlying miRNA processing and regulation. Specifically, emerging studies have shown that m^6^A modification and its regulatory proteins involve in the production of mature miRNAs, which in turn affect the level of m^6^A modification [58]. 

### 3.1. m^6^A Modifications Involves in miRNA Generation and Function

Current studies have indicated that m^6^A modification is involved in the generation process of miRNAs, thus affecting the level of mature miRNAs. Published results in the journal *Nature* in 2015 revealed that decreasing the level of m^6^A modification on pri-miRNAs by knocking down METTL3 expression could inhibit the binding of DGCR8 to pri-miRNAs, which led to about 70% miRNAs being downregulated by at least 30% [57]. Up to now, the mechanism according to which reduction of m^6^A modification on pri-miRNAs inhibits the maturation of miRNAs in a DGCR8-dependent manner has been found in different diseases. For example, catalyzed by over-expressed METTL3, high m^6^A modification can promote the maturity of miR-25 and miR-25-3p by strengthening the combination of DGCR8 and pri-miR-25 in pancreatic duct epithelial cells, and this may provoke malignant phenotype of pancreatic cancer cells [59]. The reduction level of m^6^A modification mediated by low expression of METTL3 and METTL14 makes the weaker recognition of pri-miR-126 by DGCR8, which hinders the maturation of miR-126, thereby activating the PI3K/AKT/mTOR pathway to promote the proliferation and activation of fibroblasts. Moreover, METTL3-dependent m^6^A was involved in the DGCR8-mediated maturation of pri-miR-126 in endometriosis development [60]. In addition, the interaction of METTL3 and DGCR8 positively modulates the biogenesis process of miR-873-5p, miR-365-3p and miR-221/222 in an m^6^A-dependent manner in different pathological processes, and as the simplest for specific miRNA, miR-873-5p participated in fighting colistin induced oxidative stress and apoptosis in kidney injury [61], miR-365-3p regulated chronic inflammatory pain induced by Complete Freund’s Adjuvant in the spinal cord [62], and miR-221/222 negatively mediate the PTEN expression, thus leading to the proliferation of bladder cancer cells [63]. In addition, cigarette smoke can stimulate the production of excess mature miRNA-93 in bronchial epithelial cells via enhanced m^6^A modification, which was mediated by overexpressed METTL3 [64]. METTL3 also plays a major catalytic role in m^6^A modification in unilateral ureteral obstruction mice and drove obstructive renal fibrosis development by promoting miR-21-5p maturation [65]. In addition, it has been indicated that silencing of METTL3 expression can elevate the levels of pri-miR-663 and m^6^A methylation-modified pri-miR-663, which resulted in suppressing of miR-663 maturation process in A549 and PC9LC cells [66]. In a manner similar to METTL3, the METTL14-mediated m^6^A marks also enhanced the recognition of pri-miR-126 by DGCR8, thus subsequently processing to mature miRNA-126, which is involved in hepatocellular carcinoma metastasis [60]. Different from the above mechanisms, METTL3 induced upregulation of miR-143-3p mostly depends on the shear effect of Dicer on pre-miR-143-3p in lung cancer cells [67]. Moreover, as in bone marrow-derived mesenchymal stem cells, METTL3 also methylate pre-miR-320, on which m^6^A modification is a key factor that is recognized and decayed by YTHDF2 [68]. METTL3 promoted the transition from pri-miR-1246 to mature miR-1246, of which upregulation can significantly enhance the metastasis ability of colorectal cancer cells [69]. METTL3-mediated m^6^A modification also promotes the expressions of 9 miRNAs, including miR-106b, miR-18a, miR-18b, miR-3607, miR-432, miR-30a, miR-320b, miR-320d and miR-320e, and bioinformatics analysis has shown that these miRNAs are involved in regulating signaling pathways closely related to malignant transformation induced by arsenite [70]. In addition, four miRNAs (miR-130a-3p, miR-130b-3p, miR-106b-5p and miR-301a-3p) are all related to short overall survival of kidney renal clear cell carcinoma patients and have significantly negative correlation with METTL14 mRNA [71]. Up to now, the importance of methyltransferases-catalyzed m^6^A modification on pri-miRNAs has been widely recognized, and a variety of methyltransferase components can affect the generation and function of miRNAs.

Besides methyltransferases, m^6^A demethylases and methyl-binding proteins are also involved in miRNA biogenesis. The earliest study showed that knocking down the FTO expression significantly increased levels of 42 miRNAs and decreased levels of 9 miRNAs [72]. A subsequent study reported that FTO regulates cell migration and invasion in breast cancer cells by inhibiting miR-181b-3p [73]. Moreover, FTO has been well evidenced to promoted bladder cancer cell proliferation via the FTO/miR-576/CDK6 pathways [74]. ALKBH5 inhibits tumor growth and metastasis by inhibiting miR-107/LATS2 mediated YAP activity in non-small cell lung cancer [75]. Peng et al. indicated that ALKBH5, the most potent member related to patient outcomes and to suppressing esophageal cancer malignancy in cell and animal models, demethylated pri-miR-194-2 and inhibited miR-194-2 biogenesis through an m^6^A/DGCR8-dependent manner [76]. Interestingly, in human non-small cell lung cancer cells, the depletion of ALKBH5 did not change the miR-21-5p level but altered the m^6^A abundance on miR-21-5p, thereby changing the miR-21-5p silencing potency towards its target mRNAs, which finally impaired the proliferation and motility of human non-small cell lung cancer cells [77]. In addition, ALKBH5 demethylated pri-miR-320a-3p, thus blocking DGCR8 from interacting with pri-miR-320a-3p and leading to mature process blockage of pri-miR-320a-3p in silica-inhaled mouse lung tissues [78]. 

In addition to being directly affected by m^6^A methyltransferases and demethylases, miRNA generation is also regulated by m^6^A methyl binding proteins. YTHDC1, a well-known m^6^A reader, facilitated the biogenesis of mature miR-30d via m^6^A-mediated regulation of mRNA stability. Furthermore, miR-30d represses pancreatic tumor genesis via suppressing aerobic glycolysis [79]. m^6^A reader protein HNRNPA2B1 also binds to a subset of m^6^A-modified pri-miRNA transcripts, thus interacting with DGCR8 and promoting primary miRNA processing, and depletion of HNRNPA2B1 caused a reduction in the levels of 61 miRNAs in HEK293 cells. Moreover, transiently overexpressed (5.4-fold) HNRNPA2B1 in MCF-7 cells led to significant alteration of more than 100 miRNAs, which regulate TGFβ and Notch signaling pathways according to MetaCore Enrichment analysis [80,81,82]. Yi et al. have reported that miR-185 transfer from vascular smooth muscle cells to endothelial cells is controlled by HNRNPA2B1 [83], but the role of m^6^A modification in this mediate process needs to be further explored. In addition, HNRNPA2B1 reads the m^6^A site on pri-miR-106b or pri-Let-7b to facilitate the maturing of miR-106b-5p or Let-7b in the lung cancer cells [84,85]. Another m^6^A binding protein, IGF2BP1, promotes serum response factor expression in an m^6^A-dependent manner by impairing the miRNA-directed downregulation of the *serum response factor (SRF)* mRNA in cancer cells [86]. In addition to regulating the generation process of miRNAs, m^6^A modification can directly modify mature miRNAs and affect their stability and degradation [72]. Of note, m^6^A modification on the *E2F transcription factor 3* (*E2F3*) mRNA was required for the interaction between miR-660 and *E2F3* mRNA in gastric cancer, indicating that m^6^A also affects the function of miRNA apart from participating in its production process [87]. 

In a word, emerging studies have identified the roles of m^6^A modification during the processing and maturation of miRNAs, which will surely provide good candidate targets for miRNA intervention. Although mechanisms underlying m^6^A modification affecting miRNA generation and function are diverse and complex, the general mechanisms are similar. We drew a schematic diagram, which take miR-30d [79], miR-21-5p [65] and pre-miR-25 [59] as examples, to show the specific mechanisms of m^6^A regulatory proteins regulating miRNAs (Figure 2). Additionally, the regulatory mechanisms of m^6^A modification and its regulatory proteins on miRNAs in different tissues or cells are summarized and shown in Table 1.

### 3.2. miRNAs Regulate the m^6^A Modification 

Since miRNAs affect the protein level through interacting with mRNAs and m^6^A modification is a dynamic reversible methylation [88], it is rationality that miRNAs are involved in the regulation of m^6^A modification by affecting the regulatory proteins. At present, several studies have shown that miRNAs regulate m^6^A modification via sequence pairing of mRNAs of methyltransferases, demethylases, and methyl-binding proteins in various tissues [89]. In detail, METTL3 was identified as the direct target of miR-1269b [90] and miR-338-5p [91], thus inhibiting gastric cancer development. miR-33a is capable of reducing the METTL3 expression at both mRNA and protein levels, thus affecting proliferation, survival and invasion of non-small cell lung cancer [92]. Moreover, miR-600 can attenuate METTL3 expression and restrain the migration and proliferation of lung cancer cells [93]. Similarly, the down regulation of miR-524-5p also up-regulates the expression of METTL3 in non-small cell lung cancer cells [94]. miR-4429 targeted and repressed METTL3 to inhibit m^6^A-mediated stabilization of SEC62, a component belonging to tetrameric Sec62/Sec63-subcomplex of Sec-complex, thus hindering proliferation and encouraging apoptosis in gastric cancer cells [95]. Moreover, Cai et al. concluded that mammalian hepatitis B X-interacting protein (HBXIP) suppresses miRNA let-7g, thus up-regulating METTL3, which in turn promotes the expression of HBXIP through m^6^A modification, leading to stimulation or proliferation of breast cancer cells [96]. As an independent prognostic factor in hepatoblastoma patients, METTL3 was identified as a direct target of miR-186, of which low level led to high expression of METTL3, thus significantly inhibiting the proliferation, migration and invasion of hepatoblastoma cells [97]. Moreover, miR-320d has been evidenced to target METTL3, thus affecting KIF3C expression through changing m^6^A modification on KIF3C mRNA in prostate cancer cells [98]. Under the treatment of Mono-(2-ethylhexyl)phthalate (MEHP), miRNAs such as miR-16-1-3p, miR-101a-3p, miR-362-5p, miR-501-5p, miR-532-3p and miR-542-3p are dramatically activated in murine macrophage Raw 264.7 cells, and these miRNAs are all predicted to regulate METTL14, thus promoting m^6^A modification in *Scavenger Receptor B type 1* (*SR-B1*) mRNA [99]. Cui et al. reported that miR-193a-3p directly targets ALKBH5 to inhibit the growth and promote the apoptosis of glioma cells by suppressing the AKT2 pathway both in vitro and in vivo [100]. Interestingly, circGPR137B acted as a sponge for miR-4739 to up-regulate its target FTO, which mediated m^6^A demethylation of circGPR137B and promoted its expression, thus finally forming a feedback loop comprising circGPR137B/miR-4739/FTO axis and affecting the hepatocellular carcinoma cells [101]. Results from Yang et al. indicated that imiR-155 directly targets FTO to negatively regulate its expression and increase m^6^A level in renal clear cell carcinoma cells. Regarding specific mechanisms, miR-155 is directly bound to the 3′-UTR of FTO mRNA and reduced FTO protein levels [102].

The methyl binding proteins of m^6^A modification are also directly targeted by miRNAs. In detail, Zheng et al. reported that miRNA-421-3p targets YTHDF1 to inhibit p65 mRNA translation, thus preventing inflammatory response in cerebral ischemia/reperfusion injury [103]. miR-376c also has been indicated to negatively modulate YTHDF1 expression in non-small cell lung cancer cells [104]. Negative correlations between the miR-145 level and YTHDF2 mRNA expression were observed in hepatocellular carcinoma [105] and epithelial ovarian cancer cells [106], and further detecting results showed that miR-145 decreased the luciferase activities of 3′-UTR of *YTHDF2* mRNA, implicating that YTHDF2 is the direct target gene of miR-145 [105,106]. In addition, *YTHDF2* mRNA is also regulated by miRNA-495 in prostate cancer cells [107] and miR-6125 in colorectal cancer cells [108]. Bioinformatics analysis from Hao et al. literature revealed IGF2BP1 as the putative target of miR-670, of which mimics and inhibitors were microinjected into parthenogenetic activation embryos, thus confirming these findings [109]. IGF2BP2, another m^6^A methyl binding protein, is highly expressed in thyroid cancer cells and identified as a target of miR-204 [110]. In addition, inhibition of miR-133b also resulted in the up regulation of IGF2BP2 in colorectal cancer cells [111]. Different from the above-mentioned mechanisms where miRNAs regulated m^6^A modification, results from Chen et al. indicated that overexpressing dicer increased the m^6^A modification level, and this was not achieved by alternating the quantity of m^6^A methyltransferases or demethylases in mouse embryonic fibroblasts. Further experiments showed that miRNAs regulate activity and location of METTL3, which subsequently modulate m^6^A modification and impede the reprogramming of mouse embryonic fibroblasts to pluripotent stem cells [112]. 

In a word, miRNAs can influence m^6^A modification by regulating the regulatory proteins of m^6^A and ultimately participate in a variety of pathological and physiological processes. We drew a schematic diagram, in which we take METTL3 [93], FTO [113] and YTHDF2 [108] as examples, to show the miRNAs involving in regulation of m^6^A modification and its biological effect (Figure 3). In addition, the effects of miRNAs on m^6^A regulatory proteins in different tissues or cells are summarized and shown in Table 2.

As mentioned above, m^6^A modification is regulated by different m^6^A regulatory proteins in a variety of diseases by promoting biosynthesis of miRNAs, and miRNA regulates the biological functions of m^6^A regulatory proteins. Based on this interplay, we summarized the change trends and regulation relationships of m^6^A regulatory proteins and miRNAs in tissues or cells during the occurrence and development of different diseases, as shown in Figure 4. 

## 4. Conclusions and Prospect

M^6^A modification on miRNAs has been shown to control corresponding miRNA generation and functions, thereby regulating a variety of physiological and pathological events, including proliferation, metastasis, differentiation, apoptosis and homeostasis. Interestingly, miRNAs have the potential to regulate m^6^A modifications on different RNAs through methyltransferases, demethylases and methyl-binding proteins, thereby affecting gene expression or protein function. The interaction between m^6^A modification and miRNAs provides a new direction for exploring the potential regulatory mechanisms of gene expression in different diseases. More importantly, results of studies show that surrounding RNA modifications are increasing year by year due to crucial breakthroughs in NGS-based whole-transcriptome mapping methods [114,115]. Emerging published studies have revealed a dual role for m^6^A modification in driving chromatin dynamics and transcriptional outputs [49,116]. These latest and exciting findings will help to elucidate the biological relevance of m^6^A modifications of chromatin-associated RNAs in mammalian cells in the future [117]. Nevertheless, present understanding of mutual regulatory mechanisms between m^6^A modifications and miRNAs may be only the tip of the iceberg due to the diversity of miRNA types and regulatory proteins of m^6^A modification. The interplay mechanisms between m^6^A modification and miRNAs remain to be clarified. Notably, in the field of environment and health, only sporadic studies have indicated that m^6^A modification is involved in the toxic damage caused by environmental pollutants. Given the critical roles of m^6^A modification and miRNAs in disease, further study of m^6^A and miRNAs and their mutual regulating relationship in environment-related diseases will be worthy of further discussion. 

## Figures and Tables

**Figure 1 ijms-24-00773-f001:**
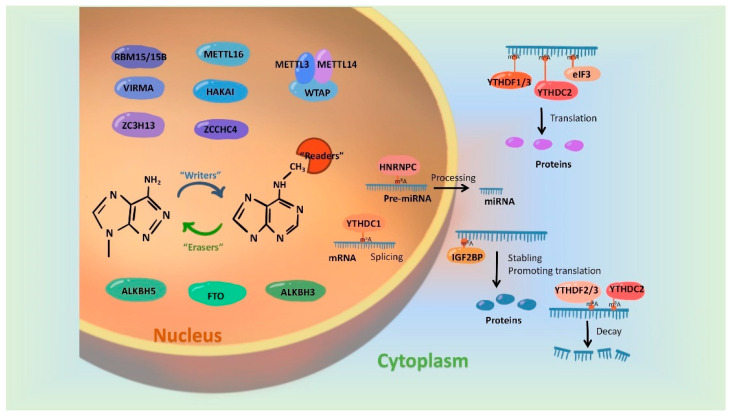
Regulatory proteins involved in mediating the m^6^A modification. m^6^A modification is synergistically catalyzed by methyltransferases (Writers), demethylases (Erasers) and methyl-binding proteins (Readers). The formation of m^6^A was initially catalyzed by a group of complexes, in which METTL3 and METTL 14 act as the active center of methyltransferases and WTAP, RBM15/15B, VIRMA and ZC3H13 play the part of adaptor proteins. In addition, METL16, HAKAI and ZCCHC4 were evidenced to be independently catalyzed the formation of m^6^A modification. FTO, ALKBH5 and ALKBH3 have been identified as m^6^A demethylases and they remove m^6^A marks on RNAs in an independent manner. Readers, involving biological functions by recognizing m^6^A modifications, include the YTH family, HNRNP family, IGFBPP1/2/3, EIF3 and ABCF1.

**Figure 2 ijms-24-00773-f002:**
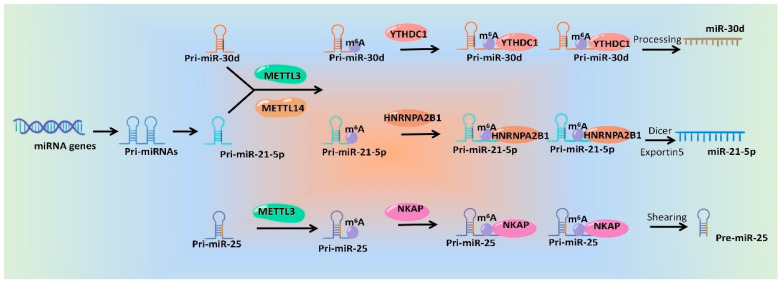
Schematic diagram of the mechanisms of m^6^A modification and its regulatory proteins regulating miRNA production.

**Figure 3 ijms-24-00773-f003:**
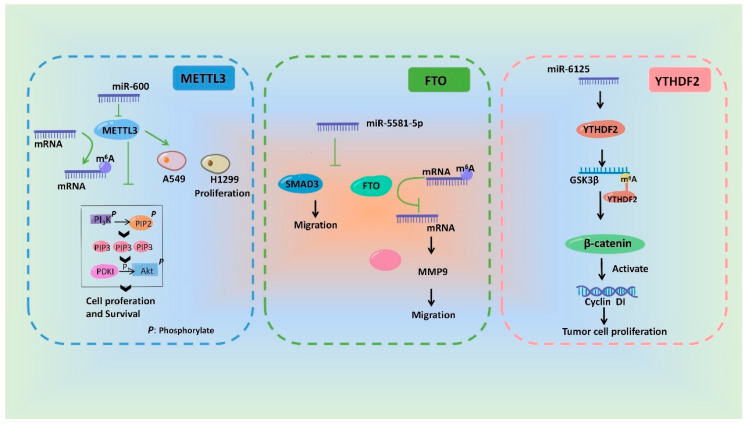
Schematic diagram of the mechanisms showing that m^6^A modification and its biological effects are regulated by miRNAs.

**Figure 4 ijms-24-00773-f004:**
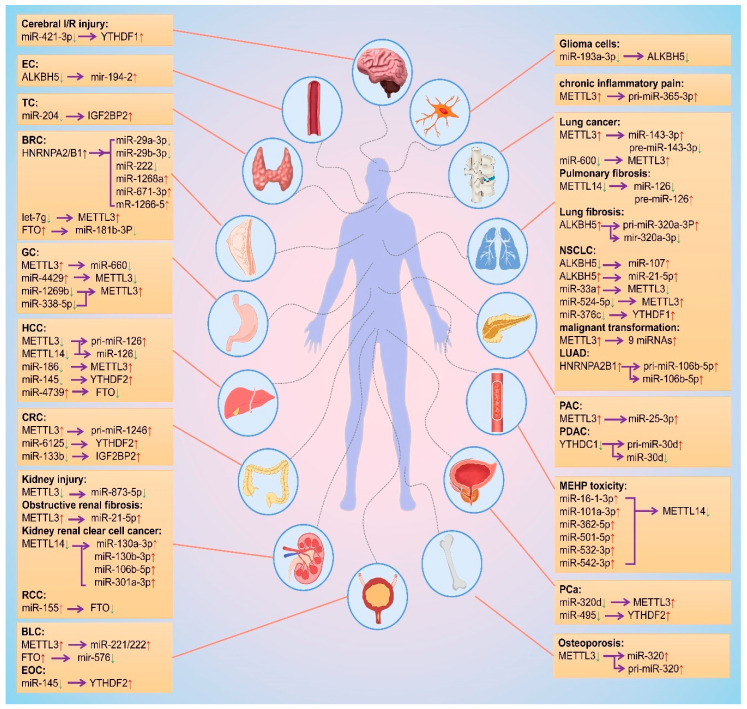
Mutual regulation between m^6^A modifications and miRNAs. The red and green arrows represent the level rise and fall, respectively. The purple arrows point to the regulated object. EC: esophageal cancer; TC: thyroid cancer; BRC: breast cancer; GC: gastric cancer; HCC: hepatocellular carcinoma; CRC: colorectal cancer; RCC: Renal cell carcinoma; BLC: bladder cancer; EOC: epithelial ovarian cancer; NSCLC: non-small cell lung cancer; LUAD: lung adenocarcinoma; PAC: pancreatic cancer; PDAC: Pancreatic ductal adenocarcinoma; PCa: prostate cancer.

**Table 1 ijms-24-00773-t001:** m^6^A modification involves in regulation of miRNA expression.

Regulatory Proteins of m^6^A	m^6^A on RNAs	miRNAs Affected by m^6^A	Mechanisms of m^6^A Mediating miRNAs	Diseases	Ref.
METTL14↓	pri-miR-126↑	miR-126↓	M^6^A modification reduced recognition of pri-miR-126 by DGCR8	Pulmonary fibrosis	[18]
METTL3↓	-	70% miRNAs↓	METTL3 catalyzed m^6^A modification on pri- miRNAs, thus promoting them for recognition and processing by DGCR8	-	[57]
METTL3↑	-	miR-25-3p↑	METTL3 catalyzed the formation of m^6^A modification on pri-miR-25 to promote its processing via NKAP	Pancreatic cancer	[59]
METTL3↓METTL14↓	pri-miR-126↑	miR-126↓	Reduction of m^6^A modification weakened recognition of pri-miR-126 by DGCR8	Hepatocellular carcinoma	[60]
METTL3↓	-	miR-873-5p↓	METTL3 interacted with DGCR8 and positively modulates maturation of miR-873-5p	Kidney injury	[61]
METTL3↑	pri-miR-365-3p↑	-	METTL3 positively affected the interaction between DGCR8 and pri-miR-365	Chronic inflammatory pain	[62]
METTL3↑	pri-miR-221/222↑	-	METTL3 positively modulated the pri-miR221/222 processing through interacting with DGCR8	Bladder cancer	[63]
METTL3↑	-	miR-21-5p↑	Up-regulation of METTL3 promoted maturation of miRNA-21-5p	Obstructive renal fibrosis	[65]
METTL3↑	pre-miR-143-3p↓	miR-143-3p↑	Knockdown of dicer can abolish METTL3-induced up-regulation of miR-143-3p	Lung cancer	[67]
METTL3↓	pre-miR-320↑	miR-320↑	METTL3 enhanced the m^6^A methylations on pre-miR-320	Osteoporosis	[68]
METTL14↓	-	miR-130a-3p↑ miR-130b-3p↑ miR-106b-5p↑ miR-301a-3p↑	MiRNA levels negative correlation with METTL14 mRNA	Kidney renal clear cell carcinoma	[71]
METTL3↑	pri-miR-1246↑	-	METTL3 methylated and promoted pri-miR-1246 maturation	Colorectal cancer	[69]
METTL3↑	Total RNAs	9 miRNAs↑	miRNA level positive correlation with METTL3 level	Malignant transformation	[70]
METTL3↑	*E2F3* mRNA	miR-660↓	m^6^A-modified E2F3 mRNA was required for the interaction between miR-660 and METTL3	Gastric cancer	[87]
ALKBH5↓	-	miR-107↑	miR-107 level negative correlation with ALKBH5	Non-small cell lung cancer	[75]
ALKBH5↓	-	miR-194-2↑	ALKBH5 inhibited miR-194-2 biogenesis through an m^6^A/DGCR8-dependent manner	Esophageal cancer malignancy	[76]
ALKBH5↑	-	miR-21-5p↑	ALKBH5 changed the miR-21-5p silencing potency by altered the m^6^A abundance	Non-small cell lung cancer cells	[77]
ALKBH5↑	Pri-miR-320a-3p↑	miR-320a-3p↓	ALKBH5 promoted lung fibrosis via the miR-320a-3p/FOXM1 axis or targeting FOXM1	Lung fibrosis	[78]
FTO↓	miRNAs	42 miRNAs↑9 miRNAs↓	FTO affected the modification of m^6^A on mature miRNA	-	[72]
FTO↑	-	miR-181b-3p↓	FTO promoting breast cancer via the FTO/miR-181b-3p/ARL5B pathway	Breast cancer	[73]
FTO↑	-	miR-576↓	FTO promoted bladder cancer cell proliferation via the FTO/miR-576/CDK6 pathways	Bladder cancer	[74]
HNRNPA2B1↑	-	miR-29a-3↓miR-29b-3↓miR-222↓ miR-1266-5↑miR-1268a↑miR-671-3p↑	HNRNPA2/B1 promoteed DROSHA processing to pre-miRNAs	Breast cancer	[81]
HNRNPA2B1↑	pri-miR-106b-5p↑	MiR-106b-5p↑	HNRNPA2B1 inhibited SFRP2 and activated Wnt-β/catenin via m^6^A-mediated miR-106b-5p processing to aggravate steaminess in lung adenocarcinoma	Lung adenocarcino-ma	[84]
IGF2BP1 ↑	-	miR-22-3p↓miR-125-5p↓miR-181-5p↓, etc.	IGF2BP1 impaired the miRNA-directed down regulation of SRF expression	Ovarian, liver and lung cancer	[86]
YTHDC1↓	Pri-miR-30d↑	miR-30d↓	YTHDC1 induced miR-30d function as a tumor suppressor gene by negatively regulating RUNX1 and its downstream glycolytic genes including HK1 and SLC2A1	Pancreatic ductal Adenocarcin-oma	[79]

Note: Sections of table with blue, green and pink background describe the process of miRNA production regulated by methyltransferases, demethylases and methyl-binding proteins, respectively. The red upward arrow represents the elevation of expression, and the green downward arrow indicates the reduction of expression.

**Table 2 ijms-24-00773-t002:** miRNAs involve in mediation the m^6^A modification.

miRNAs	Regulatory Proteins of m^6^A	Mechanisms of miRNAs Regulate m^6^A Level	Diseases	Refs.
miR-33a↑	METTL3↓	MiR-33a targeted to the 3′-UTR of METTL3 mRNA	Non-small cell lung cancer	[92]
miR-600↓	METTL3↑	MiR-600 inhibited the expression of METTL3	Lung cancer	[93]
miR-4429↑	METTL3↓	MiR-4429 targeted and inhibited METTL3 level	Gastric cancer	[95]
let-7g↓	METTL3↑	Suppressing let-7g thus up-regulating METTL3	Breast cancer	[96]
miR-186↓	METTL3↑	Low level of miR-186 led to high expression of METTL3	Hepatocellular carcinoma	[97]
miR-320d↓	METTL3↑	MiR-320d inhibits KIF3C expression by targeting METTL3	Prostate cancer	[98]
miR-1269b↓mir-338-5p↓	METTL3↑	METTL3 was the direct target of miR-1269b and miR-338-5p	Gastric cancer	[90,91]
miR-524-5P↓	METTL3↑	CircVMP1 plays an oncogenic role by targeting miR-524-5p-METTL3/SOX2 axis in A549/DDP and H1299/DDP cells.	Non-small cell lung cancer	[94]
miR-16-1-3p↑miR-101a-3p↑miR-362-5p↑miR-501-5p↑ miR-532-3p↑miR-542-3p↑	METTL14↓	MiRNAs were all predicted to regulate METTL14 mRNA	MEHP toxicity	[99]
miR-193a-3p↓	ALKBH5↓	MiR-193a-3p targeted to ALKBH5 by suppressing the AKT2 pathway	Glioma cells	[100]
miR-4739↑	FTO↓	CircGPR137B inhibits HCC tumorigenesis and metastasis through the circGPR137B/miR-4739/FTO feedback loop	Hepatocellular carcinoma	[101]
miR-155↑	FTO↓	MiR-155 directly targeted FTO and negatively regulated its expression and increased m^6^A level	Renal cell carcinoma	[102]
miR-421-3p↓	YTHDF1↑	MiR-421-3p suppresses inflammatory response during cerebral I/R injury through regulating YTHDF1/P65	Cerebral I/R injury	[103]
miR-376c↓	YTHDF1↑	YTHDF1 overexpression reversed the inhibitory role of miR-376c released by EC-Evs in NSCLC cells	Non-small cell lung cancer	[104]
miR-145↓	YTHDF2↑	MiR-145 decreased the luciferase activities of 3′-UTR of YTHDF2 mRNA	Hepatocellular carcinoma	[105]
miR-145↓	YTHDF2↑	YTHDF2 was the direct target gene of miR-145	Epithelial ovarian cancer	[106]
miR-6125↓	YTHDF2↑	MiR-6125 targeted the 3′-UTR of YTHDF2 and downregulated the YTHDF2 protein	Colorectal cancer	[108]
miR-495↓	YTHDF2↑	MiR-495 can inhibit the proliferation, invasion and migration of PCa cells by targeting YTHDF2	Prostate cancer	[107]
miR-204↓	IGF2BP2↑	IGF2BP2 was a target of miR-204.	Thyroid cancer	[110]
miR-133b↓	IGF2BP2↑	CircEZH2 interacted with IGF2BP2 and served as a sponge of miR-133b	Colorectal cancer	[111]

Note: Sections of table with blue, green and pink background describe the process of miRNA production regulated by methyltransferases, demethylases and methyl-binding proteins, respectively. The red upward arrow represents the elevation of expression, and the green downward arrow indicates the reduction of expression.

## Data Availability

Not applicable.

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
