# Peer review of "Emerging Mutual Regulatory Roles between m6A Modification and microRNAs"

_ijms, 2023, doi:10.3390/ijms24010773_

Round 1

Reviewer 1 Report

The authors Zongqin Mei et al, in their manuscript entitles " Emerging Mutual regulatory roles between m6A modification and microRNAs have attempted to discuss the mechanisms mediating M6A modification and its regulatory proteins during occurrence and development of various diseases. In recent times many such review articles have been published where the authors show the interactions between M6A and non-coding RNAs. While the authors have a commendable update on all recent publications and the updated list of microRNAs controlling the regulatory proteins of m6A in the form of their tables, there is always  some redundancy with respect to papers published in this field. With regards to making their manuscript stand out I would suggest the authors to venture into unexplored areas such as the following. I have the following major concerns and have listed them below along with some suggestions. I believe incorporating the suggestions will help to increase the overall readership and create a coherence throughout the manuscript.

a) Provide a 2 or 3 well established microRNAs controlling m6A regulation and use a schematic to show the mechanisms driving such events

b) The authors can further highlight a particular micro RNA and m6A regulation in the context of a normal physiology and a diseased context. The authors have already done this through Figure No 2 and tables1 and 2. However, a detailed explanation of the mechanism would add to the scope of this manuscript.

c) The authors can also try to highlight platforms and new NGS techniques which can be used to drive a stronger correlation between m6A and microRNAs.  Discuss how techniques like iMARGi and TT seq can help unravel novel interactions that could fill the missing pieces in the jigsaw puzzle for understanding the mechanism.

d)With respect to the language I would suggest the authors to have their paper checked by a native English speaker and revamp their manuscript.

e)With respect to the tables and the color codes used especially in table 1 and table 2 the readers can benefit from a detailed legend for the tables with regards to what the color codes stand for.

f) Additionally, the authors should italicize latin words such as in vitro and in vivo throughout their manuscript.

I strongly believe the authors can enhance on the scope of the manuscript and probably incorporate the above mentioned changes.

Thank you 

Author Response

1.Provide a 2 or 3 well established microRNAs controlling m6A regulation and use a schematic to show the mechanisms driving such events Response: We highly appreciate the reviewer’s constructive comments. According to reviewer’s suggestions, we drew a schematic diagram of the regulatory mechanisms of m6A modification involving in the regulating maturation of miRNAs (miR-30d, miR-21-5p and miR-25) and miRNAs mediating m6A modification and its methyltransferase, demethylase and methyl-binding protein. Please see these revisions in the Figure.1 and Figure.2 of the revised manuscript. 2. The authors can further highlight a particular micro RNA and m6A regulation in the context of a normal physiology and a diseased context. The authors have already done this through Figure No 2 and tables1 and 2. However, a detailed explanation of the mechanism would add to the scope of this manuscript. Response: We highly appreciate the reviewer’s careful reading and constructive comment. We have added some detail regulatory mechanisms between miRNAs and m6A modifcation interactions in the column “Mechanisms of m6A mediating miRNAs” of Table 1 and the column“Mechanisms of miRNAs regulate m6A level” of Table 2. See these revisions in Table1 and Table2 with track changes. 3.The authors can also try to highlight platforms and new NGS techniques which can be used to drive a stronger correlation between m6A and microRNAs. Discuss how techniques like iMARGi and TT seq can help unravel novel interactions that could fill the missing pieces in the jigsaw puzzle for understanding the mechanism. Response: Thanks very much for the reviewer's insightful suggestion. As pointed by reviewer, previous studies have revealed the mechanisms of action between m6A and miRNA through NGS techniques. Based on these findings, we present part of regulatory mechanisms between m6A modification and miRNAs by drawing schematic diagram. In addition, a bit of future research ideas for miRNAs and m6A modification with NGS techniques were added to the section of “Conclusion and prospect”. Please see on the Page22, Page 13 Figure.2 and Page19 Figure.3 4.With respect to the language I would suggest the authors to have their paper checked by a native English speaker and revamp their manuscript. Response 4: Thanks very much for the reviewer's kind suggestion and we are sorry that we cannot find a native English speaker to help us modify our manuscript due to various restrictions. For all that, all the authors read the manuscript and made some modifications with the help of professional grammar modification software. We have corrected the mistakes, type errors and the English grammar throughout the entirety of the manuscript. Please see these revisions in the revised manuscript with track changes. 5. With respect to the tables and the color codes used especially in table 1 and table 2 the readers can benefit from a detailed legend for the tables with regards to what the color codes stand for. Response: We highly appreciate the reviewer’s careful reading and constructive comment. According to reviewer’s suggestion, we have added detailed legend, in which we described that the red upward arrow represents an increase, while the green downward arrow represents a decrease trend for specific index value in Table 1 and Table 2. Please see these revisions in Table1 and Table2 with track changes. 6.Additionally, the authors should italicize latin words such as in vitro and in vivo throughout their manuscript. Response : We highly appreciate the reviewer’s careful reading and really apologize for this carelessness. We have italicized latin words such as in vitro and in vivo throughout our manuscript. Please see these revisions in the revised manuscript with track changes.

Reviewer 2 Report

I think that the coauthors have written an excellent review on the topic “Emerging mutual regulatory roles between m6A modification and microRNAs”. Actually it is one of the first review that summarizes the results in this exciting field providing  a prospective view of this chemical modification  and the molecules of microRNAs both in physiological and pathological contexts.  I suggest that the Editor’ service  will take care of the very many typing oversights.  The text is written in a sharp and easy manner so that the all readers can understand quickly, the readers working and not working in this field; also the tables and the figures are quite clear. This review  is a very useful product for the scientific community.   

Author Response

I think that the coauthors have written an excellent review on the topic “Emerging mutual regulatory roles between m6A modification and microRNAs”. Actually it is one of the first review that summarizes the results in this exciting field providing a prospective view of this chemical modification  and the molecules of microRNAs both in physiological and pathological contexts.  I suggest that the Editor’ service will take care of the very many typing oversights. The text is written in a sharp and easy manner so that the all readers can understand quickly, the readers working and not working in this field; also the tables and the figures are quite clear. This review  is a very useful product for the scientific community.  

Response : Thanks very much for the reviewer's nice words about our manuscript and we really apologize for the typing oversights. We have carried out careful verification and corrected the typing oversights, type errors and the English grammar throughout the entirety of the manuscript. Please see these revisions in the revised manuscript with track changes.

Round 2

Reviewer 1 Report

The Authors have incorporated the necessary changes.